# Competitive Plant-Mediated and Intraguild Predation Interactions of the Invasive *Spodoptera frugiperda* and Resident Stemborers *Busseola fusca* and *Chilo partellus* in Maize Cropping Systems in Kenya

**DOI:** 10.3390/insects13090790

**Published:** 2022-08-31

**Authors:** Johnstone Mutiso Mutua, Daniel Munyao Mutyambai, George Ochieng’ Asudi, Fathiya Khamis, Saliou Niassy, Abdul A. Jalloh, Daisy Salifu, Henlay J. O. Magara, Paul-André Calatayud, Sevgan Subramanian

**Affiliations:** 1International Centre of Insect Physiology and Ecology, Nairobi P.O. Box 30772-00100, Kenya; 2Department of Biochemistry, Microbiology and Biotechnology, Kenyatta University, Nairobi P.O. Box 43844-00100, Kenya; 3CNRS, IRD, UMR Évolution, Génomes, Comportement et Écologie, Université Paris-Saclay, 91198 Paris, France

**Keywords:** fall armyworm, ecological niche, plant-mediated competition, predation, population dynamics, stemborers

## Abstract

**Simple Summary:**

Invasion of fall armyworm led to co-inhabitation with resident stemborers in maize cropping systems in Kenya. As a result, exploitative and interference intraguild interactions occur between these lepidopteran pests. However, mechanistic insights into these interactions are poorly understood. These interspecific interactions may impact population dynamics and niche displacement of these pests in maize crops. Fall armyworm dominates resident stemborers in maize crops where they co-exist. This is due to plant-mediated and predation competitive advantages exhibited by fall armyworm over stemborers. Understanding such interactions provides crucial information for designing pest management strategies for these lepidopteran pests.

**Abstract:**

Following its recent invasion of African countries, fall armyworm (FAW), *Spodoptera frugiperda* (Lepidoptera: Noctuidae), now co-exists with resident stemborers such as *Busseola fusca* (Lepidoptera: Noctuidae) and *Chilo partellus* (Lepidoptera: Crambidae) causing severe damage to maize crops. Due to niche overlap, interspecific interactions occur among the three species, but the mechanisms and degree remain unclear. In this study, we assessed plant-mediated intraspecific and interspecific interactions, predation in laboratory and semi-field settings, and larval field occurrence of *S. frugiperda* and the two stemborer species. Larval feeding assays to evaluate competitive plant-mediated interactions demonstrated that initial *S. frugiperda* feeding negatively affected subsequent stemborer larval feeding and survival, suggesting induction of herbivore-induced mechanisms by *S. frugiperda*, which deters establishment and survival of competing species. Predation assays showed that, at different developmental larval stages, second–sixth instars of *S. frugiperda* preyed on larvae of both *B. fusca* and *C. partellus*. Predation rates of *S. frugiperda* on stemborers was significantly higher than cannibalism of *S. frugiperda* and its conspecifics (*p* < 0.001). Cannibalism of *S. frugiperda* in the presence of stemborers was significantly lower than in the presence of conspecifics (*p* = 0.04). Field surveys showed a significantly higher number of *S. frugiperda* larvae than stemborers across three altitudinally different agroecological zones (*p* < 0.001). In conclusion, this study showed that the invasive *S. frugiperda* exhibited a clear competitive advantage over resident stemborers within maize cropping systems in Kenya. Our findings reveal some of the possible mechanisms employed by *S. frugiperda* to outcompete resident stemborers and provide crucial information for developing pest management strategies for these lepidopteran pests.

## 1. Introduction

The fall armyworm, (FAW) *Spodoptera frugiperda* J.E. Smith (Lepidoptera: Noctuidae), which invaded Africa in 2016 has become a severe threat to agricultural production in the continent [1]. It has been estimated that *S. frugiperda* could cause crop losses of up to 13 billion USD per annum throughout sub-Saharan Africa (SSA), thereby threatening livelihoods of millions of poor smallholder farmers [1,2,3]. Before its invasion, stemborers, especially native *Busseola fusca* Fuller (Lepidoptera: Noctuidae) and invasive *Chilo partellus* Swinhoe (Lepidoptera: Crambidae), were the predominant lepidopteran pests on maize crops in SSA [4]. Following the *S. frugiperda* invasion, these three lepidopteran species were found to co-infest several crop plants, especially maize [5,6]. In their early larval instars, these pests have similar feeding niches, host plants, and phenological characteristics [7,8,9]. Intraspecific and interspecific interactions involving exploitative competition and predation occur [10]. These multifaceted interactions will persist as *S. frugiperda* and stemborers co-occur on several widely cultivated plants [9]. Recently, field studies revealed that the invasion of *S. frugiperda* into East African countries caused a decline in stemborer populations and a change in preference from maize to sorghum [9]. The underlying mechanisms driving such phenomena, however, were not characterized and remain poorly understood. 

Intraspecific and interspecific interactions among insects are significant biotic factors that determine the ultimate outcomes of invasive–native species interactions by influencing ecological distribution, abundance, and evolution of insects in an ecosystem [10,11]. Generally, new invasive species predominantly thrive at the expense of native and closely related species in a co-occurring ecosystem [12]. For example, a study on whitefly illustrated the competitive displacement of resident *Bemisia tabaci* (Hemiptera: Aleyrodidae) by the newly introduced *Bemisia argentifolii* Bellows and Perring (Hemiptera: Aleyrodidae) in the USA [12]. Similarly, Ekesi et al. [13] carried out a study on mango fruit fly pests and illustrated displacement of indigenous *Ceratitis cosyra* (Diptera: Tephritidae) by invasive *Bactrocera invadens* (Diptera: Tephritidae) in mango fruits. They revealed competitive superiority of invasive *B. invadens* as the primary mechanism contributing to the displacement of the native species. 

In the stemborer community, invasive *C. partellus* displaced native *B. fusca* in southern parts of South Africa, accounting for approximately 90% of the total stemborer population in sorghum plants [4]. In Kenya, *C. orichalcociliellus* (Lepidoptera: Crambidae) was dominant in maize crops in the coastal region in the 1960s. However, 10 years later, *C. orichalcociliellus* and *C*. *partellus* had equal prominence, and, two decades later, *C. partellus* dominated the coastal stemborer population in both maize and sorghum, contributing more than 80% of total stemborer infestations [14,15,16]. In Africa, a recent study illustrated how *S. frugiperda* in Uganda largely contributed to the decline in cereal stemborers despite historical reports projecting a significant increase in stemborer populations to approximately 60% of crop area in maize crops [9]. Moreover, the invasion of *S. frugiperda* in China replaced native cutworm, *Spodoptera litura* (Lepidoptera: Noctuidae)*,* and Asian corn borer, *Ostrinia furnacalis* (Lepidoptera: Crambidae)*,* to become the predominant species on maize crop within 1 year of its invasion into the country [17,18]. 

Superior competitive traits of invasive species have been associated with displacement and reduced populations of native species in a competing environment [13,14,15]. *Spodoptera frugiperda* is known for its tendency to attack a variety of crops such as maize, sorghum, rice, and tomato [19], cannibalistic behavior [20], and high reproductive rate (up to 300 eggs per oviposition event and frequent migrations of up to 1600 km in 30 h) [21]. These intrinsic characteristics of *S. frugiperda* may give it superior advantages over stemborers in a competing environment, allowing its domination in the ecosystem, and they are worth investigating in *S. frugiperda*—stemborer community interactions. 

Predation, involving consumption of heterospecific individuals, and cannibalism, entailing feeding and elimination of one’s conspecifics, are typical behaviors among insects often influenced by individual densities, food availability, and stage of development [7,20,22]. Predation and cannibalism are reported in *S. frugiperda* and *C. partellus* and may contribute to competitive advantages for *S. frugiperda*. [20,23]. Predator–prey interactions mitigate population densities of herbivore pests co-sharing an ecological niche with *S. frugiperda* in Asia [17,18]. A study by Bentivenha et al. [7] noted that predation/cannibalism frequently occurs between different larval instars and limits food and space. Predation/cannibalism can be stimulated by abiotic factors such as temperature, humidity, time, population density, and nutritional content of the plant [24,25]. Predators initiate competitive displacement and change in the population structure of prey directly through killing or indirectly by determining feeding locations [22]. Predators can affect niche utilization, food availability, and functioning of prey without immediately affecting their population structure [26,27]. Thus, *S. frugiperda,* a voracious pest with cannibalism and predation behavior, can prey and outcompete stemborers in a competing environment. Predation may, thus, be hypothesized as one of the mechanisms altering population dynamics in the *S. frugiperda*—stemborer communities. Therefore, its central role in shaping stemborer—*S. frugiperda* community needs to be clarified.

In addition to plants’ natural capacity to host folivore insects for oviposition, feeding, and regeneration, they remain critical determinants of exploitative competition among co-occurring species [28]. Plants play a role in mediating interactions of herbivore insects through constitutive and induced defense mechanisms, thereby determining their feeding, survival, growth, and development [10]. Plants have developed sophisticated defense mechanisms to guard against insect herbivory [29,30]. For example, early feeding of corn earworm *Helicoverpa zea* (*Lepidoptera*: Noctuidae) in tomatoes triggers an increase in protein inhibitor and polyphenol oxidase, which have dire consequences on the growth and survival of lepidopteran leaf chewers and aphids subsequently feeding on the tomato [31,32]. *Spodoptera frugiperda*’s initial feeding could likely invoke defense mechanisms that could affect survival, growth, and development, as well as feeding preferences, of co-inhabiting stemborer species.

Although there is literature documenting the co-existence of *S. frugiperda* with resident cereal stemborers and the decline in the pest populations, as well as a shift from maize crop [5,6,9], the mechanisms and predisposing factors remain largely unknown. Additionally, considerable research has been conducted on the effects of oral herbivore secretions in inducing defense mechanisms, [30,31,33,34], but competitive plant-mediated effects in *S. frugiperda*–stemborer interactions have never been investigated. Therefore, the objectives of this study were to (i) assess plant-mediated exploitative competition among *S. frugiperda*, *B. fusca*, and *C. partellus* in maize, (ii) determine predator–prey interactions among *S. frugiperda*, *B. fusca*, and *C. partellus* larvae, and (iii) characterize population dynamics of *S. frugiperda*, *B. fusca*, and *C. partellus* larvae in maize cropping systems across three agroecological zones in Kenya.

## 2. Materials and Methods

### 2.1. Study Site 

Laboratory and semi-field experiments were carried out at Duduville Campus (1.2921° S, 36.8219° E; 1616 m above sea level) of the International Center of Insect Physiology and Ecology (*icipe*) located in Kasarani, Nairobi City, Kenya. Field surveys were conducted in three counties (Kilifi, Makueni, and Bungoma) each representing altitudinally different agroecological zones (lowlands, midlands, and highlands, respectively). Kilifi County, a coastal lowland region, is located at 3.5107° S, 39.9093° E. Its altitude ranges from 0 to 900 m above sea level, and it is characterized by a hot and wet climatic condition throughout the year. It experiences a mean annual rainfall of over 1200 mm with mean annual temperature ranging from 30 to 34 °C. Makueni County, a mid-altitudinal region is located at 2.2559° S, 37.8937° E, and its altitude averages at 949.77 m above sea level. It is a dry transitional zone with bimodal rainfall of an annual average of 600 mm and mean annual temperature of 24.19 °C. Bungoma County is a highland region located at 0.5695° N, 34.5584° E, with a mean altitude of 1441.27 m above sea level. It lies within the areas of Kenya with high agricultural potential with maize as the main crop. It experiences rainfall throughout the year with a mean average of 1102 mm and a mean minimum temperature of 22.02 °C. 

### 2.2. Insects

Fall armyworm and stemborer (*B. fusca* and *C. partellus*) moths were reared in 80 × 50 × 70 cm oviposition cages at the Animal Rearing and Containment Unit (ARCU) at *icipe*, Duduville campus, Nairobi, Kenya. Both stemborers and fall armyworm moths were provided with maize plants separately for them to oviposit on. Eggs were collected and placed in insect rearing plastic containers for the eggs to hatch. Day-old neonates were used in all experiments which required neonates. To get older larvae for use in experiments which required different stages of larvae, neonates were reared on an artificial diet in a plastic jar (1000 mL steel-infused lids to allow air-flow) as described in Onyango and Ochieng‘-Odero [35] under optimum conditions at 27 ± 2 °C, 75% ± 5% relative humidity (RH), and L12:D12 photoperiod. Second-generation insects were used in all experiments and mixed with field-collected colonies every 3 months to prevent genetic degradation and maintain their biological characteristics. 

### 2.3. Plants

Maize seeds (SC Duma 43) were obtained from Simlaw seeds Ltd., Nairobi, Kenya. Each SC Duma 43 seed was sown in a plastic pot with soil and manure in a ratio of 2:1 in a greenhouse (28 ± 2 °C during the day, 19 ± 2 °C at night, and L12:D12 photoperiod) at *icipe.* Before planting maize seeds, pots were cleaned to eliminate cross-contamination. The maize seedlings were placed 60 cm apart and watered once daily (0.2 L) until they were 3 weeks old for use in all the experiments. This is the age commonly preferred by stemborers and *S. frugiperda* females for oviposition in maize fields [36].

### 2.4. Competitive Plant-Mediated Interactions

Experiment 1: Effects of initial feeding on subsequent larval feeding.

To evaluate the extent of larval feeding of *S. frugiperda*, *B*. *fusca*, and *C. partellus* larvae on maize plants previously exposed to intra- and inter-species interactions, 3 week old maize plants were each exposed to 10 neonates of each of the three lepidopteran herbivores separately and allowed to feed on the maize plant for 24 h. Afterward, three leaf discs of 2.0 cm diameter were cut from the newly formed leaves of each maize plant previously exposed to the three different herbivore pests. Three leaf discs were also cut from newly formed leaves of undamaged maize plants that had no prior exposure to any herbivore to act as controls. Newly formed leaves contain high levels of defense compounds in maize [37]. The leaf discs were then placed in 30 mL plastic cups containing agar medium (Technical Agar #3) to maintain the physiological status of the leaf discs. Ten naïve neonates each of *S. frugiperda*, *B. fusca*, and *C. partellus* were each placed into the 30 mL cup containing maize leaf discs. Afterward, the cups were tightly sealed with the cup lid and parafilm paper to prevent the neonates from escaping. A slit was made using a sharp scalpel at the cup lid to allow air circulation. The neonates were allowed to feed on the leaf discs for 24 h. Twelve unique treatments were established involving *S. frugiperda* on leaves initially fed on by *B. fusca, C. partellus*, and *S.frugiperda*, *B. fusca* on leaves initially fed on by *S. frugiperda*, *C. partellus*, and *B. fusca*, *C. partellus* on leaves initially fed on by *S. frugiperda*, *B. fusca,* and *C. partellus*, and all three lepidopterans on undamaged maize leaves. Each treatment was replicated 14 times. The area eaten on each leaf disc was measured using ImageJ software [38].

Experiment 2: Effects of herbivore-damaged leaves on larval development

This experiment was conducted under semi-field conditions in a greenhouse to investigate the fitness cost of herbivore-induced plant mechanisms on lepidopteran larval performance. On a 3 week old maize plant, 20 neonates (10 each of a different species) were placed on undamaged maize a whorl using a fine camel-hair brush and allowed to feed for 15 days. To act as control, 20 neonates of the same species were also separately placed on undamaged maize whorl using a fine camel-hair brush and allowed to feed for 15 days. The experiment had six treatments *S. frugiperda* + *B. fusca*, *S. frugiperda* + *C. partellus*, *B. fusca* + *C. partellus*, and *S. frugiperda*, *B. fusca*, and *C. partellus* alone on undamaged maize plant. Each treatment was replicated eight times. Maize plants were placed on metal stands raised above the ground and smeared with sticky glue (tanglefoot) to prevent ants and other predators from climbing up the stands to access infested maize plants. Maize plants were watered daily with 0.2 L of water poured at the base of the maize plant into the pot to prevent any interference with the feeding larvae in the whorl region. Survival rates, larval weight, larval length, and stage of development of each insect herbivore were recorded after 15 days. 

### 2.5. Predator–Prey Interactions

Experiment 1: Larval stage-dependent assay

To determine predator–prey interactions between *S. frugiperda* and stemborers, predation and cannibalism were investigated on maize plants (food abundance) and Petri dishes (limited food resources). For larval stage-dependent assays, first, second, third, fourth, fifth, and sixth instars of *S. frugiperda* were separately placed together with first, second, third, fourth, fifth, and sixth instars of stemborers on a whorl of maize plant placed in a 80 cm × 40 cm × 40cm cage. The first treatment was conducted by placing one sixth-instar larva of *S. frugiperda* and one larva from each of the six instars of *B. fusca* on a maize whorl separately. This was repeated with one sixth instar of *S. frugiperda* against one first, second, third, fourth, fifth, and sixth instar of *C. partellus* on maize whorl. This experiment was repeated as described above with fifth, fourth, third, second, and first instars of *S. frugiperda* against each larva of first, second, third, fourth, fifth, and sixth instars of *B*. *fusca* and *C. partellus* separately on maize whorl. In addition, the above experiment was carried out in Petri dishes with limited food resources (one 2.0 cm diameter maize leaf piece placed at the center of a Petri dish) and space to compare the level of predation in both maize whorls and Petri dishes. This established 18 treatments of all larval development stages, i.e., first–sixth instars of *S. frugiperda* against first–sixth instars of *B. fusca* and *C. partellus* on both maize whorl and Petri-dish. Control experiments were conducted in the absence of competing lepidopteran species. Each experiment was replicated eight times. The number of surviving larvae of each insect herbivore species was recorded after 24 h. Percentage predation and cannibalism was calculated as the number of missing larvae divided by the total number of larvae introduced into the treatment multiplied by 100.

Experiment 2: Density-dependent assay

Comparative studies were carried out to evaluate the strength of intraspecific (cannibalism) and interspecific (predation) interactions. Third-instar larvae of *S. frugiperda*, *B. fusca,* and *C. partellus* were chosen because, in the larval stage-dependent assay described above, *S. frugiperda* exhibited high levels of cannibalism and predation in the third instar onward. The third instar of *B. fusca* and *C. partellus* began boring into stems [39], thus reducing direct interactions with *S. frugiperda* in this density dependent assay. Eight pairs of *S. frugiperda–C. partellus*, *S. frugiperda–B. fusca*, and *C. partellus–B. fusca* were introduced on maize whorl in the greenhouse. This was repeated by reducing the number to two pairs and one pair and allowing them to feed and interact for 24 h on the maize whorl. The above experiment was repeated in the absence of competing species where 16, four, and two larvae of each of *S. frugiperda, B. fusca*, and *C. partellus* were introduced on the maize whorl and allowed to interact among conspecifics as control. Each treatment was repeated eight times. After 24 h of feeding and interaction on the maize whorl, plants were carefully dissected to remove surviving larvae of each insect herbivore species. The same assay was repeated in a Petri-dish set-up instead of a maize whorl. One maize leaf disc of 2.0 cm diameter was cut and placed at the center of the Petri-dish to act as a limited food source. The number of live larvae was recorded after 24 h. Predation and cannibalism were determined as the existing number of larvae at the end of the experiment (after 24 h) and expressed as the percentage of larvae initially introduced on the maize whorl/Petri dish.

### 2.6. Field Occurrence of Spodoptera frugiperda and Stemborers

To characterize the population dynamics of *S. frugiperda* and resident predominant stemborers, *B. fusca* and *C. partellus*, in maize cropping systems in Kenya, we carried out field larval surveys of the three lepidopterans in coastal lowland zone (Kilifi County), mid-altitude zone (Makueni County), and high-altitude highland zone (Bungoma County) of Kenya. These regions represented the zonation of Kenya based on altitude where each zone experiences different climatic conditions. In each zone, 10 maize farms with actively growing maize plants were selected for survey. To avoid concentrating on one area of the county, the surveyed maize fields were 15 km apart. The minimum size of the maize farm to be surveyed was set at 0.25 ha. In each farm, a total of 50 maize plants at different phenology (V3 and R1) determined using V notation maize phenology nomenclature [40] were randomly selected using the ‘W pattern approach’ [41] for visual inspection of the presence of *S. frugiperda,*
*C. partellus*, and *B. fusca* larvae. Maize plants in the two outermost rows of each farm were not selected for sampling. Scouting for larvae entailed first observing larvae on maize parts without causing major plant vibrations because larvae are sensitive to the vibrations and quickly drop to the ground as a means of escaping from enemies [42]. Afterward, leaves, whorls, ears, and stems were thoroughly checked for any larvae that could be hiding inside the plant parts. A sharp scalpel blade was used to open the whorl region and the stem where tunneling holes were found to collect the larvae. The number of larvae of fall armyworm and stemborers was recorded separately. Fall armyworm and stemborer abundance was recorded as the total number of larvae in the farm (50 plants). 

### 2.7. Data Analysis

Data were tested for normality using Shapiro–Wilk test. One-way analysis of variance (ANOVA) was used to analyze data on plant-mediated responses on larval feeding and predation rates. For larval feeding, initial feeding by each of the three herbivore species was treated as a fixed effect while subsequent herbivore feeding was treated as a random effect and the area consumed by larvae upon subsequent feeding as the response. For predator–prey interaction data, predator (*S. frugiperda)* was treated as a fixed effect while different larval stages of prey (stemborers) were treated as a random effect with predation rate as the response. Field occurrences of *S. frugiperda, B. fusca*, and *C. partellus* were analyzed using a generalized linear model (GLM)–quasi-Poisson distribution since it was not normal. Tukey HSD was used for mean separation. Student’s *t*-test was used to analyze differences in larval survival rates, weights, and lengths of *S. frugiperda, B. fusca*, and *C. partellus* due to co-infestation on the same maize plant. Data comparing predation and cannibalism, as well as predation on maize and Petri dish experiments, were also analyzed using Student’s *t*-test. All statistical analyses were performed using R software R i386 4.1.0 [43] with α set at 0.05.

## 3. Results

### 3.1. Competitive Plant-Mediated Effects

Experiment 1: Effects of initial feeding on subsequent larval feeding 

Initial exposure of *S. frugiperda* neonates resulted in significantly lower subsequent leaf feeding by both conspecific and interspecific larvae of *B. fusca* and *C. partellus* compared to plants initially exposed to *B. fusca* and *C. partellus* larvae and undamaged control plants. Significant differences in leaf area eaten were observed when *S. frugiperda* neonates fed on maize plants initially fed on by *S. frugiperda*, *B. fusca,* and *C. partellus*, as well as undamaged maize plants (*F*_3, 52_ = 6.38, *p* < 0.001) (Figure 1). *Spodoptera frugiperda* neonates consumed significantly more leaf area in maize plants initially exposed to *B. fusca* (*p* = 0.003) and *C. partellus* (*p* = 0.001) neonates than those initially exposed to *S. frugiperda* neonates (Figure 1). However, the leaf area consumed by *S. frugiperda* neonates was not statistically different among plants initially exposed to *B. fusca* (*p* = 0.92) and *C. partellus* (*p* = 0.71) compared to undamaged maize plants. Similarly, there were no significant differences when *S. frugiperda* fed on plants initially exposed to *B. fusca* compared to *C. partellus* (*p* = 0.92; Figure 1). *Spodoptera frugiperda* neonates also consumed significantly more leaf area from the undamaged plants than those plants initially exposed to *S. frugiperda* (*p* = 0.002) (Figure 1). 

Similarly, significant differences were observed when naïve neonates of *C. partellus* fed on plants initially exposed to *S. frugiperda*, *B. fusca*, and *C. partellus*, as well as undamaged plants (*F*_3, 52_ = 8.326, *p* < 0.001) (Figure 1). *Chilo partellus* larvae consumed less leaf area of plants initially exposed to *B. fusca* (*p* = 0.07) and *C. partellus* (*p* = 0.03) compared to undamaged plants and consumed the least when feeding on leaf tissue from maize plants initially fed on by *S. frugiperda* (*p* < 0.001) (Figure 1). There was a significant difference in the leaf area when *C. partellus* fed on plant initially exposed to *S. frugiperda* than *C. partellus* (*p =* 0.006) and *B. fusca* (*p* = 0.07). However, there was no significant difference when *C. partellus* fed on plants initially exposed to *B. fusca* compared to those exposed to *C. partellus* (*p* = 0.91) (Figure 1).

When neonates of *B. fusca* were exposed to maize plants initially fed on by *S. frugiperda*, *B. fusca*, and *C. partellus*, as well as undamaged maize plants, there was a significant difference in the area fed (*F*_3, 52_ = 10.10, *p* < 0.001) (Figure 1). *Busseola fusca* neonates fed on significantly less leaf area of plants initially exposed to *B. fusca* (*p* = 0.04) and *C. partellus* (*p* = 0.02) compared to undamaged plants and the least when exposed to maize plants initially fed on by *S. frugiperda* (*p* < 0.001) (Figure 1). A significant difference in leaf area consumed was observed when *B. fusca* fed on maize plant that was initially exposed to *S. frugiperda* compared to *B. fusca* (*p* = 0.002) and *C. partellus* (*p* = 0.03). However, there was no significant difference in leaf area consumed when *B. fusca* fed on maize plants initially exposed to *B. fusca* compared to *C. partellus* (*p* = 0.89) (Figure 1). 

In addition, there were significant differences in leaf area consumed when *S. frugiperda*, *B. fusca*, and *C. partellus* fed on undamaged maize plants (*F*_2, 39_ = 53.10, *p* < 0.001). *Busseola fusca* and *C. partellus* fed on significantly less leaf tissue than *S. frugiperda* on undamaged plants (*p* < 0.001 and *p* < 0.001, respectively). There was no significant difference in leaf area consumed by *C. partellus* and *B. fusca* neonates when exposed to undamaged maize plants (*p* = 0.98).

Experiment 2: Effects of herbivore-damaged leaves on larval development

When 10 neonates of *S. frugiperda* were exposed together with either 10 neonates of *B. fusca* or *C. partellus* on maize plant and allowed to feed for 15 days, larval survival and growth and development of *S. frugiperda* neonates were undeterred. However, fitness indices of *B. fusca* and C. *partellus* neonates were significantly reduced (Table 1). Larval survival rates, weight, and length of *B. fusca* neonates were significantly lower upon co-inhabitation with *S. frugiperda* on maize plant (*p* < 0.001, *p* = 0.01, and *p* < 0.001 respectively). Similarly, larval survival rates, weight, and length of *C. partellus* neonates declined significantly upon co-infestation with *S. frugiperda* on maize plant (*p* < 0.001 for larval survival, weight, and length). Larval survival, weight, and length of *S. frugiperda* when co-existing with *B. fusca* were not significantly different compared to its infestation in maize alone (*p* = 0.47, *p* = 0.75, and *p* = 0.60, respectively). Similarly, larval survival rates, weight, and length of *S. frugiperda* when co-existing with *C. partellus* did not change significantly compared to when it existed alone in maize plants (*p* = 0.66, *p* = 0.91, and *p* = 0.66, respectively) (Table 1). We observed that the survival rates and larval growth parameters of *B. fusca* and *C. Partellus* larvae when co-inhabited on maize plants did not differ statistically. Survival rate, larval weight, and length of *B. fusca* when co-existing with *C. partellus* was not statistically different (*p* = 0.12, *p* = 0.17, and *p* = 0.19 respectively) (Table 1). Similarly, survival rate, larval weight, and length of *C. partellus* when co-existing with *B. fusca* was not statistically different (*p* = 0.12, *p* = 0.27, and *p* = 0.14 respectively) (Table 1). After 15 days of feeding on maize plant, the most occurring larval instar of the recovered larvae differed between *S. frugiperda* larvae and those of *B. fusca* and *C. partellus*. Most of the *S. frugiperda* larvae were at fourth instar whether co-inhabiting maize plant with conspecifics or interspecific larvae. Conversely, most of the larvae of *B. fusca* and *C. partellus* were at second instar whether co-inhabiting maize plant with conspecifics or interspecific larvae (Table 1).

### 3.2. Predator–Prey Interactions

Experiment 1: Larval stage-dependent assay

When *S. frugiperda*, *B. fusca,* and *C*. *partellus* larvae were placed together on maize whorl, second–sixth instars of *S. frugiperda* preyed on both *C. partellus* and *B. fusca* (Figure 2). Predation was highest when late instars (fourth–sixth) of *S. frugiperda* co-inhabited with *B. fusca* and *C*. *partellus*, ranging from 75.0% to 82.5%. However, there was a decline in predation rates down to 62.5% when these late instars of *S. frugiperda* co-existed with early instars (first and second) of *B. fusca* and *C*. *partellus* on maize whorl. Predation differences were observed in the third instar of FAW with the lowest predation of 12.5% and highest of 75% on both *B. fusca* and *C*. *partellus*. The second instars of *S. frugiperda* expressed the lowest predation rates on early instars (first and second) and no predation on late instars (fourth–sixth) of *B. fusca* and *C. partellus*. We did not observe predation behavior by first instars of *S. frugiperda* on all instars of *B. fusca* and *C*. *partellus*, nor was there evidence of stemborers preying on *S. frugiperda*.

On the other hand, there was a high predation rate by late *S. frugiperda* on *C. partellus* and *B. fusca* in Petri dish experiments (Figure 3). Contrary to predation rates on maize plants, late *S. frugiperda* instars had higher predation rates on early stemborer instars. When late instars (fourth–sixth) of *S. frugiperda* were placed together with early instars (first and second) of *B. fusca* and *C*. *partellus* in a Petri dish, predation frequencies were highest (100%) (Figure 3), as opposed to predation rates on maize plants. Predation rates were augmented with increased *S. frugiperda* larval age and declined with stemborers’ larval age increase. As observed on maize plants, there was no predation of sixth instar of *B. fusca* and *C. partellus* by second instars of *S. frugiperda*. Similarly, we did not observe any predation by first instars of *S. frugiperda* on all stemborer instars in the Petri dish trial. There were no significant differences between predation on maize plants and Petri dish except for early instars of *S. frugiperda* against late instars of stemborers (Table 2).

No cannibalism was observed in stemborers (*B. fusca* and *C. partellus*) on maize plant and Petri dish trials. Conversely, second–sixth instars of *S. frugiperda* preyed on their younger conspecific instars on maize plant and Petri dish trials (Figure 4). When the same larval instars were placed on maize whorls or in Petri dishes, no cannibalism was observed except for the second and third instars of *S. frugiperda* (Figure 4).

Experiment 2: Density-dependent assay

At high densities (16 larvae), the level of predation by *S. frugiperda* on stemborers and cannibalism of *S. frugiperda* on its conspecifics was high (Table 3). At low densities (four larvae), the level of *S. frugiperda* predation on *B. fusca* and *C. partellus,* as wellas cannibalism on its conspecifics, was low. Increasing the number of larvae to eight larvae increased predation and cannibalism. At all densities, predation of *S. frugiperda* on both *B. fusca* and *C. partellus* was higher than cannibalism of its conspecifics (*p* < 0.001). There was no cannibalism detected when two larvae of *S. frugiperda* co-existed with two larvae of either *B. fusca* or *C. partellus*. However, as the densities increased to eight larvae, cannibalism of *S. frugiperda* in the presence of *B. fusca* and *C. partellus* differed significantly compared to cannibalism in the absence of stemborers (*p* = 0.04 and 0.01, respectively). As the density increased to 16 larvae, significant differences of cannibalism in the presence of *B. fusca* and *C. partellus* and absence of *B. fusca* and *C. partellus* increased further (*p* < 0.001 and *p* < 0.001, respectively). There was no cannibalism among *B. fusca* and *C. partellus* under all densities. Similar results were observed when density-dependent predation and cannibalism were performed on Petri dishes with limited food resources and space (Table 3).

### 3.3. Co-Occurrence of Spodoptera frugiperda and Stemborers in Maize Fields

Field studies recorded significantly more larvae of *S. frugiperda* than those of stemborers (*B. fusca* and *C. partellus)* on maize plants across the altitudinal zones (*p* < 0.001) (Figure 5). At high-elevation highland maize fields of Bungoma, 60% of the total recorded larvae were *S. frugiperda*, while 30% and 15% were *B. fusca* and *C. partellus,* respectively. In the mid-altitude zone of Makueni, 50%, 24%, and 26% of recorded larvae in maize fields belonged to *S. frugiperda, B. fusca*, and *C. partellus,* respectively. In the coastal lowland zone of Kilifi, 42%, 12%, and 36% of the total recorded larvae belonged to *S. frugiperda, B. fusca*, and *C. partellus,* respectively. 

## 4. Discussion

The present study provided empirical evidence of competitive plant-mediated and predation interactions among *S. frugiperda* and resident stemborers *B. fusca* and *C. partellus* under both laboratory and field settings. The invasion of *S. frugiperda* in Kenya in 2016 [1] raised uncertainties regarding its competitive associations with resident stemborers, especially the predominant *B. fusca* and *C. partellus*, having implications for the natural and agricultural systems. Due to the widespread similarities and sympatric occurrences of *S. frugiperda* and cereal stemborers on maize crops [5,6], niche overlap and competitive interactions are likely to occur between these herbivore pests. Interspecific competitions are known to shape population structure, distribution patterns, and biodiversity of insect species in ecosystems [10,11,24]. Consequently, the outlined exploitative (plant-mediated) and interference (predation) interspecific competitive interaction mechanisms between fall armyworm and resident stemborers are likely to shape population structure and distribution in maize cropping systems where they co-occur.

Our larval feeding assays showed differences in subsequent feeding rates upon initial exposure to competing species. Initial exposure of maize plant to *S. frugiperda* neonates adversely affected subsequent feeding of *B. fusca* and *C. partellus* neonates, while *B. fusca* and *C. partellus* initial feeding exposure had no effects on subsequent larval feeding of *S. frugiperda*. Herbivory plants are known to employ a wide range of inducible defense mechanisms [30,44,45,46]. These induced defense mechanisms make plants unpalatable for subsequent herbivore feeding [28]. Thus, herbivores often alter their preferences on plants depending on the level of the defense mechanisms. The observed larval feeding differences could be attributed to the levels of defense induction by fall armyworm and stemborer herbivores upon initial feeding of the maize plant. Notably, both the fall armyworm and the stemborers ate less area on the maize plants that they had initially fed on compared to the undamaged maize plants, an indication that the damaged maize plants had reduced palatability following initial damage. The negative feeding effects on stemborers on maize plants initially fed on by *S. frugiperda* could be associated with their inability to tolerate or sequester defense mechanisms triggered by *S. frugiperda* initial feeding. *Spodoptera frugiperda* subsequent feeding was undeterred by initial feeding of *B. fusca* and *C. partellus*, and it consumed significantly more leaf area than the maize plant which had been exposed to conspecific larvae. The increased feeding rate by *S. frugiperda* on leaf discs initially fed by *B. fusca* and *C. partellus* may indicate inability of the two stemborers to induce sufficient defense mechanisms that could deter subsequent *S. frugiperda* feeding. Additionally, this could be ascribed to the ability of *S. frugiperda* to routinely detoxify and tolerate plant defense metabolites in plants where the levels are not highly induced as fall armyworm larvae are known to sequester and detoxify plant defense metabolites [33,37,47]. The higher feeding rates of *S. frugiperda* than *B. fusca* and *C. partellus* on undamaged maize leaves could reflect its voracious characteristics of higher food consumption, which often results in faster growth and inflicts severe damage to host plants compared to the stemborers. Although beyond the scope of the current study, quantifying gene expression and subsequent defense metabolites expressed by maize following initial feeding by these herbivores will provide a clear explanation of the observed phenomenon.

Co-infestation of *S. frugiperda* and stemborers on maize plants over 15 days revealed that *S. frugiperda* larval growth was two times faster than that of stemborers. *Spodoptera frugiperda* larvae attained higher growth indices (larval weight, length, and survival) than stemborers, indicating its propensity to thrive over the other species in a competing environment. The relatively faster rate of development and higher survival indices of *S. frugiperda* compared to *B. fusca* and *C. partellus* when co-existing may essentially contribute to the displacement mechanisms. The faster growth of *S. frugiperda* than stemborers indicates that, if *S. frugiperda* infests maize plants at the same time or earlier than stemborers, it will extensively feed and induce negative plant-mediated effects for the stemborers, thus affecting their survival. Indeed, reduced survival and growth indices are indicators of poor insect competitors and the negative effects of plant defense on insect herbivores [28,29]. The faster growth of *S. frugiperda* further presents higher chances of preying on stemborers since our predation bioassays showed a high rate of predation by late instars of *S. frugiperda*. Furthermore, low survival chances and growth indices of the stemborers in the presence of *S. frugiperda* imply that stemborers suffered greater mortalities and side-effects than *S. frugiperda* when co-inhabiting maize plants. These results suggest that when both species co-occur on a young vegetative maize plant, there is a higher probability that *S. frugiperda* will have developed further than both *B. fusca* and *C. partellus*. Similar observations have also been reported when *S. frugiperda* co-occurs with other lepidopteran pests such as *Helicoperva zea*, *Ostrinia furnacalis*, and *S. litura*, competitively displacing them from the maize plants [17,18,48]. A multi-pest species systems dynamic model predicted dominance of *S. frugiperda* and *C. partellus* over *B. fusca* and *Sesamia calamistis* Hampson (Lepidoptera: Noctuidae) without extinction in maize fields in Kenya [49]. Thus, inferior ecological interactions of *B. fusca* and *C. partellus* when co-existing with *S. frugiperda* may be associated with their displacement from maize plant to other host plants such as sorghum as observed by Hailu et al. [9], who showed a decline in stemborers in maize plants and an increase in sorghum following *S. frugiperda* invasion in Uganda. 

Results from predator–prey interactions demonstrated intense detrimental consequences of *S. frugiperda* on stemborers through predation interaction. *Spodoptera frugiperda* preyed upon stemborers across all larval stages and variable densities regardless of food abundance or limitation, demonstrating its potential to outcompete and destabilize the stemborer population in the ecosystem. As early as second instars, *S. frugiperda* larvae preyed upon *B*. *fusca* and *C. partellus* under laboratory and semi-field conditions, indicating its superior characteristics in outperforming these species in a competing environment. Predation rates of early instars of *S. frugiperda* on stemborers were low, which can be attributed to their biological characteristics not being fully developed. There were no significant differences between predation on maize plant and Petri dish, especially for late instars (fourth–sixth) of *S. frugiperda,* suggesting that predation was not exclusively dependent on food resources or artificial environment, but involved other factors such as nutritional content, territorial dominance, population densities, and change of taste preferences, which are critical determinants of predation frequencies [25,50]. Predation rates of late *S. frugiperda* on early instars of stemborers declined on maize plants but increased exponentially in Petri dishes. This varying phenomenon can be ascribed to small sizes of young cohorts of stemborers and large surface area of maize plants, allowing stemborers to hide from *S. frugiperda* predation. These low predation rates on early instars of stemborers present ecological implications of *S. frugiperda*’s inability to entirely displace stemborers from maize cropping systems. Often, as stemborer larvae mature, they burrow into the stems where they feed to pupation [39], while *S. frugiperda* largely depends on maize whorl where they cause pronounced damage [51]. These differences in feeding locations present niche tradeoffs that could create possibilities of continual co-existence of *S. frugiperda* and stemborers on maize plants and provide a mechanical escape to total elimination from maize cropping systems.

Density-dependent assays demonstrated that the relative strength of interspecific interactions (predation) overwhelmingly exceeded intraspecific interaction (cannibalism) among *S. frugiperda* and stemborers. *Spodoptera frugiperda* targeted stemborers first before initiating cannibalism. In part, this could be attributed to the dominance of defensive and foraging movements of *S. frugiperda* when competing with other species for food resources and space [18]. Thus, such selective preferences of *S. frugiperda* on heterospecific individuals (stemborers) compared to their conspecifics could likely contribute to the decreased incidences of stemborers on maize plants where these lepidopterans co-exist. Despite high interspecific interactions (predation) of *S. frugiperda* with stemborers, intraspecific competition (cannibalism) was evident among the *S. frugiperda* across different larval stages and population densities. Levels of cannibalism between different larval instars of *S. frugiperda* were higher than within the same larval instars. Similar observations have been reported by other studies [20,52]. This forms self-regulation mechanisms of *S. frugiperda* populations, thereby explaining the ecological phenomena of finding a small number of *S. frugiperda* larvae on the same maize plants, especially the late instars. Although previous reports documented increased levels of cannibalism by *S. frugiperda* and its potential to shape its population structure [7,18,20], intriguingly, our study revealed a significant decrease in cannibalism rates in the presence of interspecific stemborers.

Competitive interspecies interactions can be a crucial determinant of insect species’ population abundance and geographic distribution [11]. Prior to *S. frugiperda* invasion, stemborers *B. fusca* and *C. partellus* were the abundant and most destructive lepidopteran pests of maize crops in Kenya, affecting all developmental stages of maize [4]. Field surveys carried out in this study observed a shift in the dominance of these pests on maize plants. The numbers of *B*. *fusca* and *C. partellus* were significantly lower than *S. frugiperda.* In Kilifi, Makueni, and Bungoma, the number of *S*. *frugiperda* larvae outnumbered *B. fusca and C. partellus*, demonstrating competitive isolation of stemborers and perhaps partial displacement by *S. frugiperda* in maize cropping systems in Kenya. 

Further monitoring of these pests to see how the population shifts in maize crops compared to historical occurrence data is of paramount importance and can inform pest management strategies for these herbivore pests. Current findings on the field co-occurrence of these pests underscore the role of plant-mediated and predation competitive interactions in shaping these occurrences. Thus, the shift of stemborers from maize crop to sorghum, as noted by Hailu et al. [9], could present a refugium for stemborers to evade plant-mediated and predation interaction mechanisms observed in maize plants. Such interactions have been shown to shape population of other lepidopterans. Previous field surveys performed in Kenya illustrated how *C. partellus* outnumbered and dislocated *C. orichalcociliellus* in maize crop in the coastal region within three decades of its invasion into the country [14,15,16]. The fast rate of larval development of *C. partellus* gave it a competitive advantage over the slow-developing *C. orichalcociliellus* [15]. Therefore, considering the mechanisms underlying species displacement and population dynamics in an ecosystem [13,16,53], the intrinsic behavioral and biological characteristics of *S. frugiperda* explored in this study are likely contributing factors to the reduced population of *B. fusca* and *C. partellus* and high *S. frugiperda* incidences in the three agroecological zones in Kenya.

In conclusion, the current study demonstrated that, regardless of whether the interaction was tested under laboratory or field conditions, both *B. fusca* and *C. partellus* are severely impacted by *S. frugiperda* when these lepidopteran herbivores co-occur on plants. Additionally, *S. frugiperda* was shown to dominate both *B. fusca* and *C. partellus* in maize cropping systems in Kenya across different agroecological zones. 

## Figures and Tables

**Figure 1 insects-13-00790-f001:**
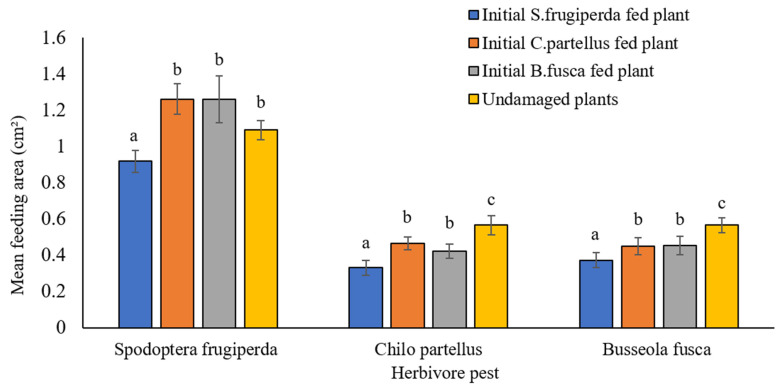
Competitive plant-mediated interactions on larval maize leaf feeding rate of *Spodoptera frugiperda*, *Chilo partellus*, and *Busseola fusca* on initially herbivore-damaged and undamaged plants. Different letters above clustered columns indicate statistically significant differences in larval feeding on herbivore-damaged and undamaged plants (one-way ANOVA; *p* < 0.05).

**Figure 2 insects-13-00790-f002:**
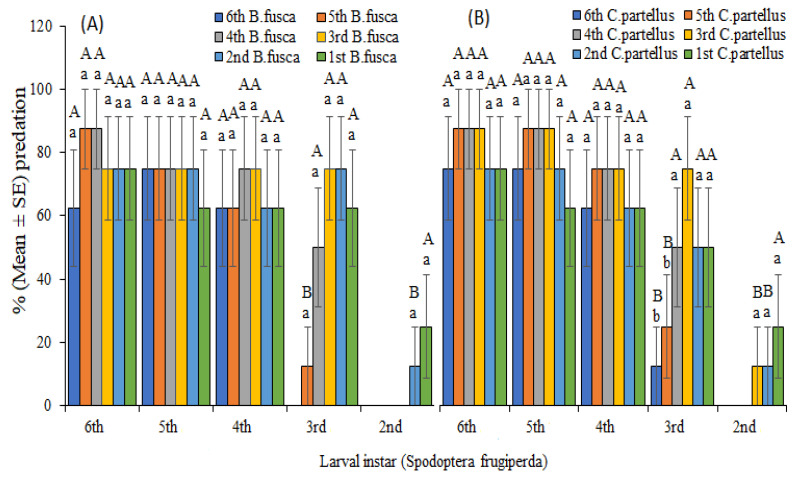
Predation rate (mean ± SE) of different larval instars of *Spodoptera frugiperda* on (**A**) *Busseola fusca* and (**B**) *Chilo partellus* in maize plant experiment after 24 h feeding. Different uppercase letters above the error bars indicate significant differences in the predation rate of given larval instars of *Spodoptera frugiperda* across larval stages of stemborers, while different lowercase letters above error bars indicate significant differences in the predation rate of each larval development stage of *Spodoptera frugiperda* on all stemborer instars.

**Figure 3 insects-13-00790-f003:**
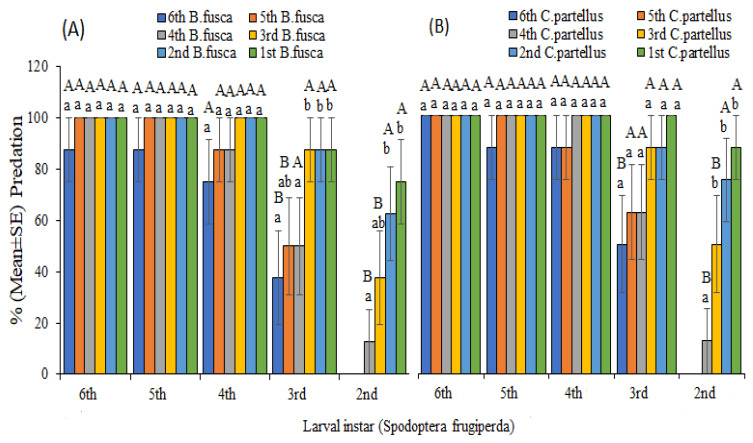
Predation rate (mean ± SE) of different larval instars of *Spodoptera frugiperda* on (**A**) *Busseola fusca* and (**B**) *Chilo partellus* in Petri dish experiment after 24 h feeding. Different uppercase letters above the error bars indicate significant differences in the predation rate of given larval instars of *Spodoptera frugiperda* across larval stages of stemborers, while different lowercase letters above error bars indicate significant differences in the predation rate of each larval development stage of *Spodoptera frugiperda* on all stemborer instars.

**Figure 4 insects-13-00790-f004:**
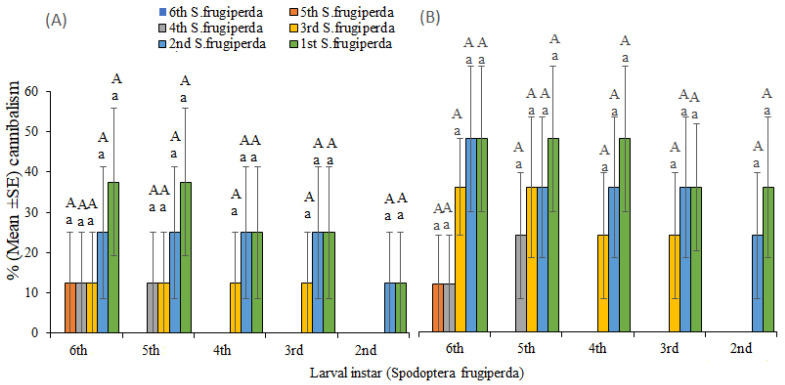
Cannibalism rate (mean ± SE) of different larval instars of *Spodoptera frugiperda* on given instars of the conspecifics on (**A**) maize plant and (**B**) Petri dish after 24 h feeding. Different lowercase letters above error bars reflect significant differences in cannibalism rate of each larval development stage of *Spodoptera frugiperda* on their conspecifics, and different uppercase letters indicate significant differences of larval stages of *Spodoptera frugiperda* across different instars of their conspecifics.

**Figure 5 insects-13-00790-f005:**
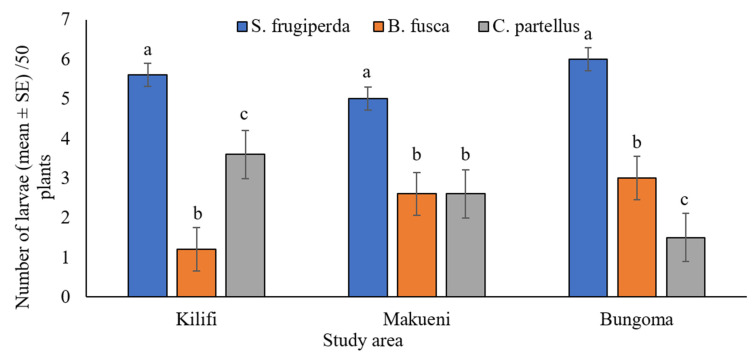
Total number (mean ± SE) of *Spodoptera frugiperda*, *Busseola fusca*, and *Chilo partellus* larvae found infesting maize fields across three agroecological zones in Kenya. Different lowercase letters above error bars reflect significant differences in the number of larvae of lepidopteran species per 50 maize plants in the study areas.

**Table 1 insects-13-00790-t001:** Percentage survival, mean larval weight, length and frequent instar when *Spodoptera frugiperda, Busseola fusca*, and *Chilo partellus* co-existed together on maize whorls for 15 days in greenhouse.

Combinations	Survival (%)	*p*-Value	Weight (mg)	*p*-Value	Length (cm)	*p*-Value
*S.f* parameters co-inhabiting with:						
*B.f*	18.75 ± 2.95	0.47	221.00 ± 55.54	0.75	3.30± 0.41	0.60
*S.f*	22.50 ± 4.12		207.40 ± 111.29		2.89 ± 0.64	
*S.f* parameters co-inhabiting with:						
*C.p*	20.00 ± 3.78	0.66	194.28 ± 44.42	0.91	3.30 ± 0.63	0.66
*S.f*	22.50 ± 4.12		207.40 ± 111.29		2.89 ± 0.64	
*B.f* parameters co-inhabiting with:						
*S.f*	7.50 ± 3.13	<0.001	1.49 ± 0.74	0.01	0.32 ± 0.15	<0.001
*B.f*	61.25 ± 7.43		3.06 ± 3.99		2.85 ± 0.50	
*B.f* parameters co-inhabiting with:						
*C.p*	11.43 ± 3.40	0.12	2.83 ± 1.00	0.17	1.20 ± 0.25	0.19
*B.f*	20.00 ± 3.78		7.94 ± 1.79		6.74 ± 1.94	
*C.p* parameters co-inhabiting with:						
*S.f*	18.75 ± 6.92	<0.001	1.50 ± 0.58	<0.001	0.71 ± 0.26	<0.001
*C.p*	92.50 ± 2.50		13.78 ± 1.50		3.57 ± 0.19	
*C.p* parameters co-inhabiting with:						
*B.f*	21.43 ± 4.59	0.12	6.74 ± 1.94	0.27	0.99 ± 0.22	0.14
*C.p*	38.57 ± 9.11		10.11 ± 2.21		1.67 ± 0.37	

**Key:***S.f* = *Spodoptera frugiperda*, *B.f* = *Busseola fusca*, and *C.p* = *Chilo partellus*.

**Table 2 insects-13-00790-t002:** Percentage (mean ± SE) predation of *Spodoptera frugiperda* on *Busseola fusca* and *Chilo partellus* exposed together on maize plant and maize leaf in a Petri dish.

Infestation		% Predation of *S.f* on *B.f*		% Predation of *S.f* on *C.p* on	
*S.f* Instar	*B.f* and *C.p* Instars	Maize Plant	Petri Dish	Maize Plant	Petri Dish
	6th	62.50 ± 18.30 a	87.50 ± 12.50 a	75.00 ± 16.37 a	100.00 ± 0.00 a
	5th	87.50 ± 12.50 a	100.00 ± 0.00 a	87.50 ± 12.50 a	100.00 ± 0.00 a
6th	4th	87.50 ± 12.50 a	100.00 ± 0.00 a	87.50 ± 12.50 a	100.00 ± 0.00 a
	3rd	75.00 ± 16.37 a	100.00 ± 0.00 a	87.50 ± 12.50 a	100.00 ± 0.00 a
	2nd	75.00 ± 16.37 a	100.00 ± 0.00 a	75.00 ± 16.37 a	100.00 ± 0.00 a
	1st	75.00 ± 16.37 a	100.00 ± 0.00 a	75.00 ± 16.37 a	100.00 ± 0.00 a
	6th	75.00 ± 16.37 a	87.50 ± 12.50 a	75.00 ± 16.37 a	87.50 ± 12.50 a
	5th	75.00 ± 16.37 a	100.00 ± 0.00 a	87.50 ± 12.50 a	100.00 ± 0.00 a
5th	4th	75.00 ± 16.37 a	100.00 ± 0.00 a	87.50 ± 12.50 a	100.00 ± 0.00 a
	3rd	87.50 ± 12.50 a	100.00 ± 0.00 a	87.5 ± 12.50 a	100.00 ± 0.00 a
	2nd	75.00 ± 16.37 a	100.00 ± 0.00 a	75.00 ± 16.37 a	100.00 ± 0.00 a
	1st	62.50 ± 18.30 a	100.00 ± 0.00 b	62.50 ± 18.30 a	100.00 ± 0.00 b
	6th	62.50 ± 18.30 a	75.00 ± 16.37 a	62.50 ± 18.30 a	87.50 ± 12.50 a
	5th	62.50 ± 18.30 a	87.50 ± 12.50 a	75.00 ± 16.37 a	87.50 ± 12.50 a
4th	4th	75.00 ± 16.37 a	87.50 ± 12.50 a	75.00 ± 16.37 a	100.00 ± 0.00 a
	3rd	75.00 ± 16.37 a	100.00 ± 0.00 a	75.00 ± 16.37 a	100.00 ± 0.00 a
	2nd	62.50 ± 18.30 a	100.00 ± 0.00 b	62.50 ± 18.30 a	100.00 ± 0.00 b
	1st	62.50 ± 18.30 a	100.00 ± 0.00 b	62.50 ± 18.30 a	100.00 ± 0.00 b
	6th	0.00 ± 0.00 a	37.50 ± 18.30 b	12.50 ± 12.50 a	50.00 ± 18.90 a
	5th	12.50 ± 12.50 a	50.00 ± 18.90 a	25.00 ± 6.37 a	62.50 ± 18.90 a
3rd	4th	50.00 ± 18.90 a	50.00 ± 18.90 a	50.00 ± 18.90 a	62.50 ± 18.90 a
	3rd	75.00 ± 16.37 a	87.50 ± 12.50 a	75.00 ± 16.37 a	87.50 ± 12.50 a
	2nd	75.00 ± 16.37 a	87.50 ± 12.50 a	75.00 ± 16.37 a	87.50 ± 12.50 a
	1st	62.50 ± 18.30 a	87.50 ± 12.50 a	50.00 ± 18.90 a	100.00 ± 0.00 b
	6th	0.00 ± 0.00 a	0.00 ± 0.00 a	0.00 ± 0.00 a	0.00 ± 0.00 a
	5th	0.00 ± 0.00 a	0.00 ± 0.00 a	0.00 ± 0.00 a	0.00 ± 0.00 a
2nd	4th	0.00 ± 0.00 a	12.50 ± 12.50 a	0.00 ± 0.00 a	12.50 ± 12.50 a
	3rd	0.00 ± 0.00 a	37.50 ± 18.30 b	12.50 ± 12.50 a	50.00 ± 18.90 a
	2nd	12.50 ± 12.50 a	62.50 ± 18.30 b	12.50 ± 12.50 a	75.00 ± 16.37 b
	1st	25.00 ± 16.37 a	75.00 ± 16.37 b	25.00 ± 16.37 a	87.50 ± 12.50 b

**Key:***S.f* (*Spodoptera frugiperda*), *B.f (Busseola fusca),* and *C.p* (*Chilo partellus*). Different lowercase letters along the same row reflect statistically significant differences between predation of *Spodoptera frugiperda* on *Busseola fusca* and *Chilo partellus* on both maize plant and Petri dish (*p* < 0.05).

**Table 3 insects-13-00790-t003:** Percentage predation and cannibalism of *Spodoptera frugiperda* on *Busseola fusca, Chilo partellus*, and conspecifics on maize plant and maize leaf in a Petri dish.

Localities	Population Density	Predation of *S.f* on:		Cannibalism of *S.f* in Presence of:			*P* ^sb^	*P* ^sc^	*P* ^bp^	*P* ^cp^
		*B.f*	*C.p*	*S.f*	*C.p*	*B.f*				
Maize	4	56.25 ± 6.25	62.50 ± 8.18	12.50 ± 12.50	0.00 ± 0.00	0.00 ± 0.00	0.33	0.33	<0.001	<0.001
Maize	8	62.50 ± 9.45	71.88 ± 7.38	18.75 ± 6.25	3.13 ± 3.13	6.25 ± 4.09	0.04	0.01	<0.001	<0.001
Maize	16	70.31 ± 5.76	76.56 ± 3.69	29.69 ± 4.05	6.25 ± 4.09	7.81 ± 2.29	<0.001	<0.001	<0.001	<0.001
Petri dish	4	87.50 ± 8.18	93.75 ± 4.09	25.00 ± 18.30	0.00 ± 0.00	0.00 ± 0.00	0.14	0.14	<0.001	<0.001
Petri dish	8	93.75 ± 4.09	96.88 ± 3.13	34.38 ± 8.10	3.13 ± 3.13	6.25 ± 4.09	0.007	0.002	<0.001	<0.001
Petri dish	16	95.31 ± 2.29	98.44 ± 1.56	45.31 ± 3.29	21.88 ± 3.13	21.88 ± 3.13	<0.001	<0.001	<0.001	<0.001

**Key:***S.f* (*Spodoptera frugiperda*), *B.f* (*Busseola fusca*), and *C.p* (*Chilo partellus*). *P*^sb^ indicates the *p*-value for comparison of cannibalism rates of *S. frugiperda* when competing with *B. fusca* and conspecifics. *P*^sc^ refers to the *p*-value for comparison of cannibalism rates of *S. frugiperda* when competing with *C. partellus* and conspecifics. *P*^bp^ is the *p*-value for comparison of predation of S. *frugiperda* on *B. fusca* verses cannibalism of *S. frugiperda* in the presence of *B. fusca.*
*P*^cp^ is the *p*-value for comparison of predation of S. *frugiperda* on *C. partellus* verses cannibalism of *S. frugiperda* in the presence of *C. partellus.*

## Data Availability

The data that support the findings of this study are currently available from the corresponding author upon reasonable request.

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
