# Peer review of "Competitive Plant-Mediated and Intraguild Predation Interactions of the Invasive Spodoptera frugiperda and Resident Stemborers Busseola fusca and Chilo partellus in Maize Cropping Systems in Kenya"

_insects, 2022, doi:10.3390/insects13090790_

Round 1

Reviewer 1 Report

The authors have constructed a well-written manuscript addressing an imported pest, S. frugiperda, in Kenya maize production systems. These findings will improve understanding of S. frugiperda population dynamics and inform integrated pest management programs for this pest and endemic stemborer populations. 

Comments and suggestions for major revisions are listed below. The quality of tables/charts to enhance presentation/readability could be improved. Justification for data analysis methodology is needed.

Simple Summary

L23-25: The last sentence is challenging to read; suggest removing “use when.”

Abstract

L44-45: I would be specific here; replace with “…displace them within maize cropping systems in Kenya.”

L45-48: Consider replacing with, “Our findings reveal mechanisms employed by FAW to outcompete resident stemborers and provide crucial information for developing pest management strategies for these lepidopteran  pests.”

Introduction

L69: “stemborer poluations”

L80: replace “United States of America” with “USA”

L94: 60% of what? “…significant increase of stemborer populations in maize to approximately 60% of the crop area.”

L114: Replace “phenomenal” with “suitable”

L127: “…altering population dynamics in stemborer-FAW communities.”

L145: “…negative impact on subsequent visitors to the same plant.”

L150: Replace with “Although there is literature documenting the co-existence of…”

L158: Replace “FAW’s” with “FAW.” Moreover, please use consistent language throughout the manuscript (e.g., use FAW or S. frugiperda).

Methods

L252: It is not clear in this section what 1st ­– 6th instars means. Was instar randomly selected? 

L324-333: Data analysis section needs more detail. Were the fixed and random effects defined for experiments where one-way ANOVA was used? Perhaps a linear mixed model (LMM) would be most appropriate to test the significance of multiple fixed and random effects with the continuous variable in question (i.e., plant-mediated response, predation rate, field occurrence) – for example, fixed effects for species, treatment, and species*treatment interaction (or study area, species, area*species interaction), and random effects for replication. Please expand the data analysis section in the text and justify the analysis method used for each experiment in your response.

Results

L382: Figure 1 is difficult to differentiate pattern schemes. Replace patterns with colors (I strongly recommend this for all figures). Fix the legend to include all treatments.

L420: Use consistent formatting for all tables. A combination of bold and underlined headings in table columns makes the table difficult to read. I recommend using the same formatting as Table 1 for other tables.  

L527: Figures 2 and 3 need to be combined. I suggest a four-panel grid layout. All figures need to be smaller in size on the page. Use color to differentiate bars on charts. To improve readability of graphics, include a bar above bar groupings that says “ns” (not significant) or the significant p-value donated by a number or asterisk(s) – letters above graphs indicate comparisons across all treatments.

L598: Be consistent with table description formatting. Combine “Notes” and “Key” sections and remove the title “Notes/Key.”

L618: Figure 5 y-axis titled is superimposed on the labels. Please clean figure formatting for all figures to enhance readability.

Discussion

L680: “acquiring essential nutrients” seems unnecessary and makes the sentence challenging to read. Suggest omitted. 

Reviewer 2 Report

Line 30: ...teractions occur among the three species, but the mechanisms and degree remain unclear.

Line 31-32:  list what you assessed in the same order as they appear in materials and methods and results. 

Line 33:  …and the two stemborer species…

Line 34:  you evaluated competitive interactions – but the term “plant-mediated” should not be explained more clearly.  Later in the paper you discuss the role of inducible plant resistance, but here plant-mediated could mean many different things. 

Line 34:  Once you have established that FAW is the acronym for fall armyworm, then continues to use it throughout and be consistent, don’t return to alternately using S. frugiperda and FAW.

Line 37:  You did not actually investigate the colonization of plants by the lepidopterans, just the survival.

Line 39-40:  If you state that a finding was statistically significant, provide the p value, even in the abstract.

Line 43:  explain that your field surveys were conducted across three altitudinally different agroecological zones. 

Line 43:  In conclusion, this study showed – not shows – everything is written in past tense.

Line 44:  exhibited – past tense, not exhibits

Line 45:  …could displace them… this is far too speculative.  And unclear.  What is meant by displace?  Your field study showed that they are coexisting in maize even after 6 years post-invasion.  FAW and stemborers have different niches and if there is niche overlap, as in Line 29, then there needs to be a clear explanation of where and when the overlap occurs.  The stemborers must have enough FAW free space in order to continue to persist.

Line 44-47:  …exhibited a clear competitive advantage over resident stemborers, some of the possible mechanisms involved and informs potential management strategies against FAW and stem borers. 

Line 61:  Following the FAW invasion, these three species were found to co-infest several crop plants, especially maize [5,6].

Line 64:  Replace the term guild with niche.  The term guild refers to a group of unrelated insect species feeding in a similar manner on a shared resource.  The term niche is more appropriate for the way you are writing the sentence – the niche is the way in which an insect feeds.  So FAW and the stemborers have similar niches.  They are only a guild when they are feeding similarly; when the stem borers move into the stems they are no longer a guild and have different niches. 

Line 65:  Can you provide a citation for exploitive competition with invasive species?  And why do you say “likely”?  Aren’t you observing exploitive competition?  If so, then say exploitive competition has been observed instead of likely.

Line 68:  Please provide a citation for which cultivated plants they co-occur on.

Line 78:  All whiteflies are phytophagous, therefore remove the term phytophagous.

Line 79:  B. tabaci is not a native species to the US, it was also invasive.  Provide the author and (Hemiptera: Aleyrodidae) to be consistent and proper in writing scientific names.

Line 80:  specific epithets are not capitalized.  Bemisia argentifolii Bellows and Perring (Hemiptera: Aleyrodidae).

Line 82:  utilize proper scientific name format for Ceratitis cosyra and Bactrocera invadens. 

Line 84:  removal is a poor word choice to describe the results of the competitive displacement.  The flies were not removed physically.  Nor does it impart where they went.  It is better to use displacement than removal. 

In line 79 you use the term native and in line 84 you use the term indigenous.  If they are equal terms, then choose one and be consistent.  Using both causes confusion. 

Line 87:  You introduce a new species without the proper formatting.

Line 95:  crop should be plural – crops.

Line 96:  Two new species names introduced therefore they need proper formatting for first use in a paper.

Line 99:  remove ‘and’.  Superior competitive traits…

Line 100:  population should be plural – populations

Line 101:  Remove The new invasive species.  Start sentence with FAW

Line 102:  at the end, remove ‘and’ before among. 

Line 104:  high reproductive rate, up to 300 eggs per oviposition event, and frequent migrations of up to 1600 km…

Line 106:  Use FAW instead of S. frugiperda

Line 108:  investigating.  Remove “in a multifaceted FAW-stemborer….”

Line 112:  Rewrite line 112 to read as: Predation and cannibalism are reported in FAW and C. partellus and may contribute to competitive advantages for FAW.  Remove the rest of the sentence.

Line 115:  Rewrite as follows:  Predator-prey interaction mitigate population densities of herbivore pests co-sharing the ecological niche with FAW in Asia [17,18].

Line 119:  Remove “Nevertheless”

Line 121:  nutritional content of the plant?

Line 123:  Remove “Interestingly”

Line 126:  remove “conveniently”

Line 129: rewrite as:  the stemborer-FAW community and needs to be clarified.  Remove all the rest. 

Remove lines 132-134 and rewrite as follows starting on line 134:  Plants play are role in mediating interactions of herbivorous insects through constitutive and induced…[28,10].

Line 137-139 – rewrite as follows:  Plants have developed sophisticated defense mechanisms to guard against insect herbivory [29,30].

Line 140: proper formatting for new mention of Helicoverpa zea. 

Line 143:  subsequently feeding on the tomato [31,32].  Remove the rest of 143 through 146.  Start new sentence with FAW’s initial feeding…. Do not use the term exquisite.

Line 150: you state that the literature is replete, but only provide three citations.  It is better to say “Although there is literature on the co-existence of…

Line 156:  Do not use the term unravel.  Also use past tense always.  State the objectives as you’ve done starting on line 158.  Remove from 156 the sentence “This study aims…

Lines 166-183.  There are grammar and formatting issues.  The paragraph can be clearer with basic editing. 

Line 185:  stemborers should be singular or remove the term “moth”

Line 209:  You are not measuring a feed rate, but rather an amount.

Remove lines 210 and 211, start sentence with “Three week old maize plants…

Line 218:  You start the sentence with “These newly-formed leaves…”. But which set of leaves are you referring to because you have previously described two different sets of leaves. 

Line 219:  space between 30 and ml needed for consistency.  There are a lot of these typos throughout the materials and methods.  Also remove small, not needed because you’ve already quantified them as 30 ml. 

Line 222 remove “small” and so on throughout the explanation.  Just use 30 ml cup. 

Line 237:  weeks should be singular – week

Line 258: Add ‘a’ – “on a maize whorl…” Fix throughout the rest of the explanation.

Line 264:  in Petri dishes with

Line 265: center of a Petri dish

Line 269: on both maize whorls and in Petri dishes.

Line 273:  you are not measuring a rate.  The percentage of predation and cannibalism was calculated.  There is no such thing as the percentage rate. 

Line 277:  Third instar larvae of…

Line 278: rewrite as follows:  were chosen because in the larval stage-dependent assay described above, FAW exhibited high levels of cannibalism and predation beginning in the third instar onwards.  The third instars of B. fusca and C. partellus began boring into stems [39], thus reducing direct interaction with FAW larvae in this feeding experiment.

Line 296:  again, you are not measuring a rate.

Line 325:  Data were tested.  Data is plural, datum is singular.

Line 338:  …neonates resulted in significantly lower

Line 348:  B. fusca

Line 357: consumed less leaf area..

Line 358:  …undamaged plants and consumed the least when feeding on

Line 368:  fed on significantly less leaf area

Line 374: …difference in leaf area consumed when…

Line 376: replace fed with consumed as above

Fix Line 378 as above in 368

Continue to clean up results section.  The results are fine and make sense, but a few grammatical corrections will help readability and clarity.  Make sure not to refer to “rate” or “rates” as you did not measure rates.

Line 536:  in Petri-dish experiments.

Line 567:  were placed on maize whorls or in Petri-

Line 584 and 585:  not a rate

Line 623:  …predation interactions among FAW and resident stemborers B. fusca and C. partellus under laboratory and semi-field settings.

Line 630:  Intraguild is not used correctly.  Replace with Interspecific

Line 633:  but they are only intraguild until the 3rd instar when the stemborers enter the plant, correct?  Once the stemborers are in the stem feeding, is there no longer any competition?

Line 672:  Could you please cite studies showing the degree day studies/models for FAW and B. fusca and C. partellus?  That will help explain the more rapid development of FAW under the same conditions as the stemborers. 

Line 697: Give a citation number for Hailu et al.

Line 716: declined on maize plants

Line 717: exponentially in Petri-dishes

Line 722: stemborer larvae

Line 724:  replace devastation with damage

Round 2

Reviewer 1 Report

Thank you for addressing and incorporating reviewer comments – the manuscript is much improved. 

The figures are much clearer; still, I have a few additional comments for improvement and enhanced readability. Y-axes should be the same for A and B in all figures. For all figures, only one legend is required (color scheme for each respective instar). In the x-axis, the name of the species can only be listed once to improve readability (e.g., either in the top header – “A) S. frugiperda” OR below the x-axis – “Larval instar (S. frugiperda).” may need to be printed smaller to fit on the page. For some figures, the y-axis limit may need to be increased to prevent cropping of letters/bars.

I recommend another thorough read-through before final publication. I caught a few minor grammatical errors throughout (e.g., L354  – “For predator-prey interactions data, predator (S. frugiperda) was treated as a fixed effect while different larval stages of prey (stemborers) were treated as random effects with predation rate as the response.”). 

Author Response

Dear editor,

Please find response to reviewer 1 report round 2

Comment 1: Thank you for addressing and incorporating reviewer comments – the manuscript is much improved.

Response 1: Thank you for this comment and we appreciate that the manuscript is now improved. 

Comment 2: The figures are much clearer; still, I have a few additional comments for improvement and enhanced readability. Y-axes should be the same for A and B in all figures. For all figures, only one legend is required (color scheme for each respective instar). In the x-axis, the name of the species can only be listed once to improve readability (e.g., either in the top header – “A) S. frugiperda” OR below the x-axis – “Larval instar (S. frugiperda).” may need to be printed smaller to fit on the page. For some figures, the y-axis limit may need to be increased to prevent cropping of letters/bars.

Response 2: Thank you for this comment. We are glad that the figures are much clearer. We have maintained same Y-axes for A and B in the figures. We have addressed these issues in the revised figures.

Comment 3: I recommend another thorough read-through before final publication. I caught a few minor grammatical errors throughout (e.g., L354  – “For predator-prey interactions data, predator (S. frugiperda) was treated as a fixed effect while different larval stages of prey (stemborers) were treated as random effects with predation rate as the response.”). 

Response 3: Thank you for this comment. We have read through the manuscript once more and corrected grammatical errors which we could point out.

Reviewer 2 Report

The rewrite of the paper is well done.  All concerns were met. 

Author Response

Response to reviewer 2.

Thank you for your inputs and comments. We are glad the paper can now read well.